# Enhancing Bank Loyalty through Sustainable Banking Practices: The Mediating Effect of Corporate Image

Nicholas Igbudu * [ID], Zanete Garanti [ID] and Temitope Popoola [ID]

Business Administration Department, Cyprus International University, Nicosia 99258, North Cyprus, Turkey; zgaranti@ciu.edu.tr (Z.G.); 21805961@student.ciu.edu.tr (T.P.)
* Correspondence: 20154532@student.ciu.edu.tr; Tel.: +90-5428792079

**Abstract:** As the demand for a more sustainable society increases, adopting a sustainable banking approach serves as a competitive advantage for banks that are focused on attaining bank loyalty. This study revolves around understanding the role of sustainable banking practices in bank loyalty while exploring the mediating effect of corporate image on the relationship between sustainable banking practices and bank loyalty. For this study, 511 questionnaires derived from customers of the banking sector were adopted. Results from structural equation modeling showed that sustainable banking practices positively and directly affected bank loyalty and corporate image, corporate image directly and positively affected bank loyalty, and corporate image also mediated the relationship between sustainable banking practices and bank loyalty.

**Keywords:** sustainable banking; corporate image; bank loyalty

---

## 1. Introduction

The deterioration of the climate, which can be traced to unsustainable practices, calls for the development and adoption of a more sustainable approach for the daily activities of humans. A sustainable approach is considered to be one involving practices that enhance the attainment of both the present needs of humanity as well as the needs of the future [1]. This requires the development of a blueprint that ensures the attainment of these needs [2]. At an organizational level, sustainability entails being cognizant of the needs of the organization's stakeholders [3], which encompasses consciousness of organization profitability, the planet, and people [4].

In recent times, the need for sustainability has led to the development of various innovative sustainable business approaches [5–8]. To be more precise, the issue of sustainability has been viewed from the social perspective [9–11], the technological perspective [12], and the organizational perspective [1,13,14]. Furthermore, eight archetypes have been categorized into three sustainable dimensions: technological, social and organizational dimensions [15]. Technological sustainability refers to the process of adopting innovative measures that efficiently utilize energy in the maximization of materials [15,16]. Social sustainability focuses on striking a balance between the attainment of basic needs and not dilapidating the environment [10].Organizational sustainability emphasizes the need for firms to be more concerned and committed to the future well-being of the organization's environment while pursuing present goals [1].

Sustainability can be seen as a strategy needed to steer values, which has also been considered to be the yearning of clients [17]. Therefore, this calls for improvement in the sustainable performance of financial institutions. The quest for more sustainable approaches has led to the development of microfinance institutions established for the purpose of improving social performance (contrary to the

conventional motives of most financial firms [18]), which has been said to be a valuable mechanism in the attainment of social and financial performance [19]. Furthermore, improving sustainable performance requires inculcating sustainability in the guidelines of financial firms both locally and internationally [20]. In the international scene, the need for an improvement in sustainable performance has led to the development and adoption of a sustainable developmental goal (agenda 2030) by the member states of the United Nations [21] that has the sole aim of attaining social and economic growth as well as ensuring global partnership. This has been supported by powerful regional financial regulatory bodies such as the European Investment Bank (EIB) through the establishment of the "sustainable awareness bond", which is aimed at encouraging sustainable businesses and is in tune with the goals of agenda 2030.

The role of banks in the economy calls for a greater sustainable approach to be adopted by firms in the industry. Sustainable banking can be seen as a process of utilizing financial products and services in creating a prosperous environment [22]. This approach has been said to be operational in very few banks [23,24], despite the insistence from customers for more sustainable practices [25].

In spite of the increase in the relationship between customers and their banks, which has made customer loyalty a top priority by the management of most banks [26], there seems to be no visible study that explores a customer's perspective on sustainable banking and its impact on bank loyalty. Studies in this field have been focused on either delineating an ideal sustainable approach for the industry [16] or proposing sustainability as an ideal approach for enhancing performance in the banking industry [27–29].

The primary aim of this study was centered on the role of sustainable banking in bank loyalty as well as the mediating effect of corporate image on the relationship between sustainable banking practices and bank loyalty. The focus of this research was on customers of the banking sector in North Cyprus. The need for an improvement in the practices of banks on the small Mediterranean island (North Cyprus in particular) was the impetus for this research, as previous studies on customers' perceptions of the industry indicated the failure of firms of the industry to meet the expectations of customers through the services they offered [30].This has been traced to the unethical practices of banks [31], which were against the principles of sustainability. Therefore, an understanding of the customer's perspective on sustainable banking—which is considered to emanate from the broader concept of ethical banking [32]—and its relation to bank loyalty could serve as a blueprint in the eradication of unsustainable practices in the banking sector.

## 2. Theoretical Framework

*Socially Responsible Investment*

A shift in customers' interests from returns on investment to ethical standards and social and environmental commitments of firms [33] stresses the importance for the adoption of a socially responsible investment (SRI) approach by firms. SRI, otherwise known as sustainable investment, can be seen as an investment policy that is centered on the positive attainment of financial, social, and environmental fortunes.

There have been several propositions through which firms can adopt the SRI ideology. For instance, banks can offer incentives such as low interest rate on borrowed capital to partners that pursue sustainable goals, while partners with unsustainable cultures are made to pay higher interest rates [34]. Furthermore, as a means of meeting desired ethical, environmental, and social goals, firms should be expected to adopt the SRI approach by developing corporate strategies that closely engage the local community and are related to the activities of shareholders [35]. Given the above assertions, there seem to be similarities between the SRI approach and the corporate social responsibility (CSR) approach, with findings from a prior study indicating that adopting the CSR approach enhanced the values of shareholders in the long run, while firms that ignored such approaches tended to raze shareholders' values, which was capable of destroying a firm's reputation [35].

## 3. Literature Review

### 3.1. Bank Loyalty

Several studies have been undertaken to understand bank loyalty [36–38].The propositions from these studies regarding bank loyalty are indistinguishable from those of brand loyalty. Bank loyalty was considered as an incidence whereby a customer repeatedly chose a particular bank over others, which was done through an evaluative process [37]. As a means of understanding factors that influence bank loyalty, previous studies have indicated factors such as corporate image as essential [37,39], which makes corporate image an important factor for management when formulating strategies [39].

Regarding the relationship between sustainability and bank loyalty, there seem to be limited studies that have clearly linked both constructs, though it has been stated that a customer's intention to purchase a brand is greatly influenced by the credibility of that brand [40]. From this assertion, a credible bank in the eyes of customers could be one which develops and adopts sustainable approaches in the delivery of financial services, which could then be considered a foundation for bank loyalty.

### 3.2. Sustainable Banking

The intermediary role of banks is said to be significant to the economy [16,41,42]. On the other hand, adopting a sustainable banking approach is considered to be synonymous with the adoption of sustainable innovative approaches in other industries [43], which has focused on taking into consideration the desires of the organization's stakeholders as well as on ensuring a serene environment and society when proposing values [44,45]. Therefore, the core of any innovative model should revolve around the creation of values with economic, environmental, and societal benefits [46].

In relation to the principles of innovative sustainability, sustainable banking could be seen as an ideology motivated by the need for new and sustainable approaches that can be used in transforming the industry [34,47], by applying innovative technologies that aid in the efficient and effective delivery of banking services [48]. The concept of sustainability in the banking industry has also been considered to be a philanthropic act whereby banks, through their services and products, create values that protect the well-being of society through positive or ideal investment [49], which in return has been said to be of huge economic benefit to banks [28]. In another study, sustainable banking was considered to be the process of creating ethical values for stakeholders, which is also instrumental in consolidating and transforming the banking industry [50]. It could also be described as an act of developing a culture that is centered on financial, social, and environmental performance: on fostering long-term relationships with customers: on inclusive and transparent governance, and on meeting the needs of both the economy and the community [51]. This approach has not only been said to be stakeholder-oriented, it has also been recommended for adoption, when evaluating the performance of banks [27]. A more comprehensive study conceptualized a sustainable banking approach by developing eight archetypes categorized into three dimensions: technological, social, and organizational. This approach, if effectively implemented by banks, was discovered to positively affect the purchase intention of customers of the banking industry [16].

As sustainability continues to be an issue of great concern, customers tend to build a positive perception about a brand with sustainable features. This is considered to be a mechanism for enhancing the corporate image of brands [52]. As a result of this intense yearning from customers, banks are now modifying their products and services by offering packages with moderate interest rates and significant environmental benefits to society [53].This could be considered to be toeing the line of sustainability, where the customer's perception of sustainable banking has been indicated to be positive [16]. This implies that customers subscribe to sustainable banking practices, which could then serve as a yardstick for bank loyalty, if they are properly adopted by banks. It is in relation to the above assertions that we posited the following hypotheses:

**H1.** Sustainable banking positively affects corporate image;

**H2.** Sustainable banking positively affects bank loyalty.

*3.3. Corporate Image*

The need for broad research on the image of banks is said to be as important as research on the financial effectiveness of the banking industry [54]. A significant amount of studies have viewed corporate image to be stakeholders' perceptions of the features of a company [55–57]. This has been said to yield a significant influence on the behaviors of customers and the performance of firms [58]. A related study viewed corporate image as the emotional (affections) and functional (measurable tangible features) perceptions of people toward the organization [59]. Therefore, as a means of enhancing reputation, firms now resort to investing heavily in CSR activities that aid in building long-term values for stakeholders [35]. Furthermore, a similar study viewed corporate image as an appropriate measure to use in eradicating consumers' doubts about a company [60].

From the banking perspective, corporate image is seen as a visual identity, which is considered important for the retail banking sector because of the similarities in products and services rendered [61]. "Corporate behavior and corporate visual identity", are factors that build corporate image, which is determined by "dynamism, stability/credibility, client/customer service and visual identity" [37]. Furthermore, the "Corporate image of commercial banks includes dimensions related to the services offered, accessibility, corporate social responsibility, global impression, location and personnel" [62].

Managing corporate image in the banking industry requires that firms understand the target market environment and the views of stakeholders, respond promptly to negative perceptions about the bank, enhance customer satisfaction as well as that of other stakeholders, have a precise position in the market, base decisions on objective rationality, and ensure the right means of communication are adopted [54].

Regarding the relationship between corporate image and loyalty, there have been a considerable number of studies on both constructs. From a multidimensional approach, research conducted in the Netherlands, exploring the perceptions of customers indicated a positive relationship between corporate image and bank loyalty [37]. A similar study carried out in the service industry in Canada showed that the level of loyalty was hugely determined by customers' perceptions of the corporate image of a service provider [63]. A study conducted in Zimbabwe, indicated that corporate image had a direct influence on bank loyalty and mediated the relationship between service quality and bank loyalty [39]. It is based on the findings of this related literature, that we posited the following hypotheses:

**H3.** Corporate image affects bank loyalty;

**H4.** Corporate image mediates in the relationship between sustainable banking and bank loyalty.

## 4. Methods

The items used in developing the research model for this study are seen in Figure 1 and were adopted from valid studies related to our interest. The questionnaire was made up of four sections: demography, sustainable banking, corporate image, and bank loyalty. The demography section was comprised of age, gender, marital status, and educational qualifications. Ten items were adopted in measuring sustainable banking practices [16,64], five items were adopted in measuring corporate image [39], and six items were adopted in measuring bank loyalty [39].

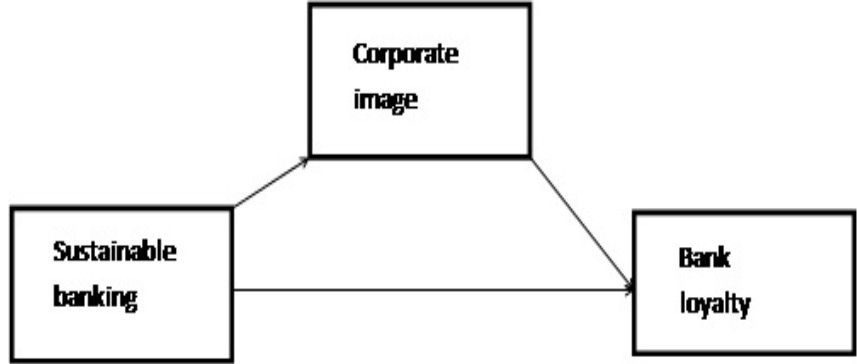

**Figure 1.** Research model.

We obtained 511 data from customers of banks in North Cyprus. With English and Turkish being the languages most spoken in the northern part of the Mediterranean island, the questionnaire items were developed in English and translated into Turkish using the back-to-back method [65]. The Turkish version of the questionnaire was reviewed by three professors in the Department of Business Administration of Cyprus International University for its acceptability and adequacy. Furthermore, the questionnaire was distributed electronically via various social media platforms by appealing and explaining to respondents about the essence of the research.

Results from Table 1 indicate that there were more male (55.6%) respondents than female respondents (44.4%). Of the respondents, 28.6% were 18–25 years of age, 25.6% were 26–35 years of age, 24.3% were 36–45 years of age, and 21.5% were 46 and above. The questionnaire indicated that 51.9% of respondents were married, 48.1% were single, and 64.2% had at least a bachelor's degree.

**Table 1.** Questionnaire items adopted for the research.

| Construct | Description | Frequency |
|:---:|:---:|:---:|
| Gender | | |
| Male | 284 | 55.6 |
| Female | 227 | 44.4 |
| Age | | |
| 18–25 | 146 | 28.6 |
| 26–35 | 131 | 25.6 |
| 36–45 | 124 | 24.3 |
| 46 and above | 110 | 21.5 |
| Education | | |
| BSc | 328 | 64.2 |
| MSc | 95 | 18.8 |
| PhD | 11 | 2.2 |
| Others | 77 | 15.1 |
| Marital Status | | |
| Married | 265 | 51.9 |
| Single | 246 | 48.1 |

The analytical techniques adopted for this research included confirmatory factor analysis (CFA) and structural equation modeling (SEM). Initially, an exploratory factor analysis (EFA) was conducted and indicated some items with standardized loading <0.50 and cross-loading of variables. Such items were eliminated (i.e., *sb1*, *sb2*, *sb3*, *sb4*, *sb5*, *sb6*, *sb7*, *bl1*, and *bl6*).

### 4.1. Measurement Validity

Values of standardized factor loading, Cronbach's alpha, composite reliability, average variance extracted (AVE), and model fit indices were used to assess convergent validity. The overall model fit was assessed for both CFA and SEM using the comparative fit index (CFI), goodness of fit index (GFI), adjusted goodness of fit index (AGFI), normal fit index (NFI), chi-squared test with degrees of freedom ($X^2$/df), root mean square error of approximation (RMSEA), and standard root mean square residual (SRMR).

The results in Table 2 show that the convergent validity for this study was acceptable. The measurement model fit indices (CFI = 0.989, GFI = 0.964, AGFI = 0.944, NFI = 0.981, $X^2$/df = 2.345, RMSEA = 0.051, and SRMR = 0.0444) were all in accordance with the recommended level [66,67]. The standardized factor loading for all items were above the minimum recommended level of 0.6 [68]. With 0.7 considered to be the minimum acceptable value [69], the Cronbach's alpha values for all constructs in this study were acceptable. Furthermore, a CR value of 0.9 for all constructs, which according to research should not be less than 0.7 [70], is acceptable.

**Table 2.** Convergent validity.

| Construct | Items | Standardized Factor Loading | Cronbach's Alpha | Composite Reliability |
|---|---|---|---|---|
| Sustainable banking | *SB8* | 0.911 | 0.943 | 0.953 |
| | *SB9* | 0.941 | | |
| | *SB10* | 0.949 | | |
| Corporate image | *CI1* | 0.884 | 0.880 | 0.873 |
| | *CI2* | 0.855 | | |
| | *CI3* | 0.860 | | |
| | *CI4* | 0.565 | | |
| | *CI5* | 0.600 | | |
| Bank trust | *BL2* | 0.953 | 0.879 | 0.967 |
| | *BL3* | 0.954 | | |
| | *BL4* | 0.970 | | |
| | *BL5* | 0.873 | | |

Note: $X^2$/df = 2.345, CFI = 0.989, GFI = 0.964, AGFI = 0.944, NFI = 0.981, RMSEA = 0.051, and SRMR = 0.044. $X^2$/df: chi-squared test with degrees of freedom; CFI: comparative fit index; GFI: goodness of fit index; AGFI: adjusted goodness of fit index; NFI: normal fit index; RMSEA: root mean square error of approximation; and SRMR: standard root mean square residual.

### 4.2. Discriminant Validity

We compared the AVE with the squared inter construct correlation (SCI), and with AVE > SCI [71,72], this indicated the presence of discriminant validity, as seen in Table 3.

**Table 3.** Discriminant validity.

| Construct | M | SD | SB | CI | BL |
|---|---|---|---|---|---|
| *SB* | 2.921 | 1.103 | 0.934 | | |
| *CI* | 2.906 | 1.099 | 0.474 | 0.766 | |
| *BL* | 2.889 | 1.100 | 0.598 | 0.433 | 0.938 |

Note: M: mean, SD: standard deviation, SB: sustainable banking, CI: corporate image, BL: bank loyalty.

### 4.3. Test of Hypotheses

According to the results as seen in Table 4, sustainable banking practices positively affected corporate image ($\beta = 0.498$, $p < 0.01$) and bank loyalty ($\beta = 0.744$, $p < 0.01$), which supported hypotheses H1 and H2. Further results indicated a positive direct effect of corporate image on bank loyalty ($\beta = 0.112$, $p < 0.01$), which supported hypothesis H3. Furthermore, results from the mediation analysis conducted indicated that corporate image significantly mediated the relationship between sustainable

banking practices and bank loyalty ($\beta = 0.055$, $p < 0.01$), which upheld hypothesis H4. Further findings indicated no significance for our control variables on the study variables.

**Table 4.** Test of hypotheses.

| Hypothesis | Path | Standardized Estimate | Remark |
|---|---|---|---|
| H1 | $SB \rightarrow CI$ | 0.498 | Supported |
| H2 | $SB \rightarrow BL$ | 0.744 | Supported |
| H3 | $CI \rightarrow BL$ | 0.112 | Supported |
| H4 | $SB \rightarrow CI \rightarrow BL$ | 0.055 | Supported |

## 5. Discussion

There are several theoretical implications that can be derived from this study. Primarily, this study investigated customers' perspectives on sustainable banking practices and their impact on bank loyalty, also exploring the mediating effect of corporate image on the relationship between sustainable banking practices and bank loyalty. This is necessary because, as stated earlier, in this emerging field of study the few empirical studies available viewed sustainable banking with regard to the most appropriate sustainable model that should be adopted [16,34], as well as individuals' perceptions of the importance and performance of sustainable practices on society [73]. Therefore, viewing sustainable banking from the relationship marketing perspective adds substance to this emerging field of research.

This study indicates that sustainable banking practices had a positive impact on bank loyalty and on the corporate image of banks, implying that adopting sustainable banking practices sends a positive signal to the bank stakeholders regarding their position and what they represent in the corporate environment. This buttresses prior findings that stressed the relationship between the former and latter variables [74].

Further findings from this study showed a positive effect of corporate image on bank loyalty. In other words, a positive perception from a bank's customer enhanced the customer's loyalty to that brand, which upholds the prior position that corporate image serves as an antecedent to bank loyalty [34].

Corporate image had a mediating role in the relationship between sustainable banking practices and bank loyalty. This serves as an important finding of this research because, to the best of our knowledge, there seems to be no study that has been able to investigate the mediating effect of corporate image on the relationship between the two variables, despite previous findings indicating an indirect impact of corporate image on customer loyalty [75].

With customer loyalty increasingly becoming a competitive factor in the acquisition of market shares, and with customers preferring that their banks adopt more ethical or sustainable approaches, an understanding of the various factors that are in tune with the customer's view on sustainable practices likely serves as a competitive advantage for any bank that is focused on acquiring and consolidating loyalty through the values they offer to the market.

Applying the right sustainable approach requires banks to move from developing strategies for economic returns only, to adopting a strategic corporate sustainability management approach through redefining strategic processes, content, and context. This creates room for the integration of sustainable values for the banks and stakeholders at large [76].

As a means of developing a more sustainable approach capable of winning bank loyalty, banks are advised to adopt approaches such as a cognitive mapping approach, which centers on gathering information from senior management or experts, which is instrumental in the development of an ideal framework or model [76]. If properly adopted, these approaches will enable banks to gather sustainable information that can be used in making sustainability visible within the corporate strategy, including the development of financial products and services and in the adoption of digital processes and a stewardship role. These are congruent with the desires of modern-day customers.

To be more precise, and as stated earlier, after the indictment of banks in North Cyprus for unethical and unsustainable practices, adopting a sustainable approach should tend to serve as redemption from the negative perception of banks on the small island. This can definitely build a positive image for any bank committed to such an approach, which in the long run could be seen as a competitive advantage in the attainment of loyal customers in the industry.

As stated earlier, the essence of this study was aimed at viewing sustainable banking from the relationship marketing perspective by understanding the role of sustainable banking practices in bank loyalty. Just like all other research, there were some limitations that could be identified for this study. First, the study was conducted on bank customers in the northern part of Cyprus, which to an extent does not represent the general view of bank customers in the world. Therefore, exploring customers' perspectives on sustainable banking practices and their impact on bank loyalty in a larger society, or in a comparative study between two or more countries, would tend to add more substance to this study.

This study combined the technological, social, and organizational dimensions of sustainability as factors for sustainable banking practices. Future research could explore other factors such as the economic dimension, which could also enrich this field of study. Furthermore, treating the three dimensions adopted for this study as separate variables may also give an in-depth view of which out of the three variables customers value the most, which could be useful to policy makers.

**Author Contributions:** All three authors collectively contributed to the writing, reviewing, and editing of this article.

**Funding:** This research received no external funding.

**Conflicts of Interest:** The authors declare no conflict of interest.

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
