# Peer review of "Enhancing Bank Loyalty through Sustainable Banking Practices: The Mediating Effect of Corporate Image"

_sustainability, doi:10.3390/su10114050_

Reviewer 1 Report

As the authors state, adopting a sustainable banking approach can represent a competitive advantage for Banks with a positive impact on customers' loyalty. Thus, by means of 511 data from customers of banks in North Cyprus, the authors explore the mediating effect of corporate image  when studying the relation between sustainable banking practices and bank loyalty. The analytical techniques adopted for this research are confirmatory factor analysis (CFA) and structural equation modeling (SEM). Results show that corporate image is seen to have a mediating role in the relationship between sustainable banking practices and bank loyalty. The paper is well structured and written; however, some minor changes would improve the quality of the study. First of all, Figure 1 looks blurred.  English language and style are fine but minor spell check is required. Last but not least, there is abundant literature on financial institutions and sustainable performance. Topics like social and environmental performance have been widely studied. Therefore, i would recommend the authors to include a couple of references like the following ones (among others) in the introduction section with a brief introductory paragraph:

Bartual Sanfeliu, C.; Cervelló Royo, R. and Moya Clemente, I. (2013) Measuring performance of social and non-profit Microfinance Institutions (MFIs): An application of multicriterion methodology."Mathematical and Computer Modelling 57.7-8 (2013): 1671-1678.

Cervelló-Royo, R., Moya-Clemente, I., & Ribes-Giner, G. (2015). Microfinance Institutions (MFIs) in Latin America: Who Should Finance the Entrepreneurial Ventures of the Less Privileged?. In New Challenges in Entrepreneurship and Finance (pp. 235-245). Springer, Cham.

Daher, L., & Le Saout, E. (2013). Microfinance and financial performance. Strategic Change, 22(1‐2), 31-45.

Oyegunle, A., & Weber, O. (2015). Development of Sustainability and Green Banking Regulations: Existing Codes and Practices.

Weber, O. (2013). Measuring the impact of socially responsible investing.

Weber, O., & Remer, S. (2011). Social banks and the future of sustainable finance. Routledge.

Weber, O. (2005). Sustainability benchmarking of European banks and financial service organizations. Corporate Social Responsibility and Environmental Management, 12(2), 73-87.

Author Response

Dear Reviewer, 

 Thank you very much for finding out time to review our article titled “Enhancing bank loyalty through sustainable banking practices: the mediating effect of corporate image”. We are grateful for your effort.

In line with your recommendations, figure 1 has been made clear; a paragraph has been added to the introduction, stressing the efforts of financial institutions on sustainable performance, and the minor spelling errors has been corrected as well. All these has been indicated in the manuscript, through the ‘’track change’’ function in Microsoft word.     

 Once again, we thank you for your guidance and recommendations.

Reviewer 2 Report

Good the intention to show that sustainable banking practices positively and directly affects bank loyalty and corporate image. I think this is very interesting for the readers.  

I would suggest enriching the introduction referring to, for example, international agreements on sustainability (Agenda 2030 and the Sustainable Development Goals). It is also important to mention that the European Investment Bank (EIB) recently has launched a fund called "Sustainable awareness bond" as a financial instrument to make a concrete contribution to the achievement of the 2030 Agenda aimed to invest in business projects related to sustainability.

Author Response

Dear Reviewer, 

Thank you very much for finding out time to review our article titled “Enhancing bank loyalty through sustainable banking practices: the mediating effect of corporate image”. We are grateful for your effort.

In line with your recommendations, a paragraph has been added to the introduction, stressing the efforts of financial institutions on sustainable performance, as well as the efforts of international and regional bodies such as the United Nations (agenda 2030), and European Investment Bank (sustainable awareness bond), in the improvement of sustainable performance. All these has been indicated in the manuscript, through the ‘’track change’’ function in Microsoft word.

Once again, we thank you for your guidance and recommendations.